



# Impact of volcanic eruptions on CMIP6 decadal predictions: A multi-model analysis

Roberto Bilbao[1], Pablo Ortega[1], Didier Swingedouw[3], Leon Hermanson[4], Panos Athanasiadis[5], Rosie Eade[4], Marion Devilliers[6], Francisco Doblas-Reyes[1,2], Nick Dunstone[4], An-Chi Ho[1], William Merryfield[7], Juliette Mignot[8], Dario Nicolì[5], Margarida Samsó[1], Reinel Sospedra-Alfonso[7], Xian Wu[9], and Stephen Yeager[10]

[1]Earth Science Department, Barcelona Supercomputing Center (BSC), Barcelona, Spain
[2]Institució Catalana de Recerca i Estudis Avançats (ICREA), Barcelona, Spain
[3]Environnements et Paléoenvironnements Océaniques et Continentaux (EPOC) Univ. Bordeaux, CNRS, Bordeaux, France
[4]Predictability Research Group, MetOffice, Exeter, UK
[5]Climate Simulations and Predictions Division, Centro Euro-Mediterraneo sui Cambiamenti Climatici (CMCC), Bologna, Italy
[6]Danish Meteorological Institute, Copenhagen, Denmark
[7]Canadian Centre for Climate Modelling and Analysis, Environment and Climate Change Canada (ECCC), Victoria, Canada
[8]Laboratoire d'Océanographie et du Climat (LOCEAN), Sorbonne Université-CNRS-IRD-MNHN, Paris, France
[9]Atmospheric and Oceanic Sciences Program, Princeton University, Princeton, USA
[10]Climate and Global Dynamics Division, National Center for Atmospheric Research (NCAR), Boulder, USA

**Correspondence:** Roberto Bilbao (roberto.bilbao@bsc.es)

**Abstract.** In recent decades three major volcanic eruptions of different intensity have occurred (Mount Agung in 1963, El Chichón in 1982 and Mount Pinatubo in 1991), with reported climate impacts on seasonal-to-decadal timescales that could have been potentially predictable with accurate and timely estimates of the associated stratospheric aerosol loads. The Decadal Climate Prediction Project component C (DCPP-C) includes a protocol to investigate the impact of such volcanic eruptions on decadal prediction. It consists in repeating the retrospective predictions that are initialised just before the last three major volcanic eruptions but without the inclusion of the volcanic forcing, which are then compared with the baseline predictions to disentangle their effect. We present the results from six CMIP6 decadal prediction systems. These systems show a strong agreement in predicting the radiative response to the volcanic eruptions, simulating a reduction in global mean top-of-atmosphere radiation fluxes, surface temperature and ocean heat content. The characteristic geographical patterns of the response are consistent across the models and share similarities across the volcanic eruptions, however some differences across models and eruptions arise due to the varying magnitude and spatiotemporal structure of the volcanic forcing. Taking advantage of the large multi-model ensemble we additionally analyse the dynamical responses in the Northern Hemisphere atmospheric circulation, in the tropical Pacific Ocean and the North Atlantic Ocean. Comparing the predicted surface temperature anomalies in the two sets of hindcasts (with and without volcanic forcing) with observations we show that including the volcanic forcing results in overall better predictions. The volcanic forcing is found to be particularly relevant for reproducing the observed SST variability in the North Atlantic Ocean following the 1991 eruption of Pinatubo, however in the tropical Pacific Ocean the predicted SST anomalies are degraded.



# 1 Introduction

Decadal climate predictions have become a major tool for forecasting the climate of the next few years out to several decades
(e.g. Hermanson et al., 2022). On these timescales, part of the predictability arises from internal variability, in particular in
the slowly evolving components of the climate system (e.g. the ocean). This predictability can be improved by initialising
the model with the observed state to put the model in phase with observed internal variability. The other main source of
predictability relates to the changes in external radiative forcings (i.e. changes in the climate system energy balance), which

can be of natural (e.g. solar irradiance and volcanic aerosols) or anthropogenic (e.g. greenhouse gas concentrations, land use
changes and anthropogenic aerosols) origin. At the global scale, most of the observed surface temperature changes can be
explained by the warming caused by the increasing atmospheric greenhouse gas concentrations, which are partly compensated
by the cooling caused by anthropogenic aerosols, and the sporadic cooling episodes that followed the major volcanic eruptions.
Hence, including the volcanic forcing and correctly simulating its impacts is a major input for reproducing the observed

climate variability in the immediate years after their occurrence. Furthermore, from a climate forecasting perspective, if a
major volcanic eruption were to occur, it would be necessary to run new forecasts including the respective forcing since the
former may significantly impact how the climate evolves in the following years.

In recent decades three major tropical volcanic eruptions have occurred: Mount Agung (1963), El Chichón (1982) and Mount
Pinatubo (1991). These eruptions of varying intensity ( 7 Tg,  8 Tg and  18 Tg of $SO_2$ respectively) had climate impacts on

seasonal-to-decadal timescales with high predictive potential (e.g. Timmreck et al., 2016; Ménégoz et al., 2018; Hermanson
et al., 2020). Explosive volcanic eruptions affect climate by injecting large quantities of sulphur dioxide (as well as other gases
like water vapour, $CO_2$ and dust) into the stratosphere, where it oxidises to form sulphate aerosols. The presence of sulphate
aerosols in the stratosphere has two main effects: (1) reflects part of the incoming solar radiation, causing a negative radiative
forcing that cools the Earth's surface, an effect that may last for several years (until the aerosols return to the surface) and

(2) absorb infrared radiation and block the outgoing longwave radiation which may lead to a local warming of the strato-
sphere (Robock, 2000). These temperature adjustments may subsequently lead to other climate impacts on seasonal-to-decadal
timescales (see Marshall et al. (2022), for a review), such as atmospheric and oceanic dynamical changes, which may modulate
climate variability.

Observational and modelling studies have shown the increased likelihood of a positive phase of the Northern Annular

Mode (NAM) or a positive North Atlantic Oscillation (NAO) like response in the first winter following the eruptions (e.g.
Robock, 2000; Christiansen, 2008). The positive NAM/NAO response has been explained by the acceleration of the Northern
Hemisphere polar vortex as a result of the anomalous equator-to-pole temperature gradient in the stratosphere, caused by the
post-volcanic stratospheric warming in the lower latitudes (e.g. Graf et al., 1993; Stenchikov et al., 2002). Such a response has
been linked to warming over the North Eurasian continent in winter, consistent with studies based on observations (e.g. Robock

and Mao, 1992; Shindell et al., 2004), although these are limited to few large volcanic eruptions. Paleoclimate studies based on





proxy reconstructions show a robust NAO response for eruptions larger than Pinatubo (e.g. Ortega et al., 2015; Michel et al., 2020). Modelling studies have shown a wide range of results. While some modelling studies have concluded that Coupled Model Intercomparison Project 5 (CMIP5) climate models might be unrealistic by underestimating the Northern Hemisphere polar vortex response (e.g. Driscoll et al., 2012; Charlton-Perez et al., 2013), Bittner et al. (2016) showed that disregarding the smaller eruptions and only including Krakatoa (1883) and Pinatubo (1991), the models on average do simulate a strengthening of the vortex. More recent studies have highlighted the need of large ensembles and/or very strong volcanic eruptions (e.g. Tambora in 1815) to detect climate signals such as the NH polar vortex strengthening or the Eurasian winter surface warming, as these can be overwhelmed by internal variability (e.g. Ménégoz et al., 2018; Polvani et al., 2019; Azoulay et al., 2021; DallaSanta and Polvani, 2022). Furthermore, it is yet unclear whether these signals might be underestimated due to the signal-to-noise ratio problem affecting the North Atlantic atmospheric circulation in the current generation of Earth system models (Scaife and Smith, 2018). In fact, Hermanson et al. (2020) show that the NAO anomaly in the first winter after the eruption is about seven times weaker in the climate predictions than in the observations, despite the stratospheric heating having the right magnitude.

Volcanic eruptions also impact the tropical Pacific ocean variability. Despite some discrepancies across studies regarding the response of El Niño Southern Oscillation (ENSO) to volcanic eruptions, most studies show an El Niño-like response in the first year following an eruption (e.g. Meehl et al., 2015; Maher et al., 2015; Swingedouw et al., 2017; McGregor et al., 2020; Hermanson et al., 2020). Three main mechanisms have been proposed that trigger changes in the ENSO state following a large tropical volcanic eruption: (1) the "ocean dynamical thermostat" mechanism (ODT) (Clement et al., 1996), (2) Maritime Continent land-ocean temperature gradient (Predybaylo et al., 2017) and (3) altered Walker circulation in response to the reduction of tropical precipitation over Africa (Khodri et al., 2017). There is yet no consensus as to which mechanism dominates. For example, Pausata et al. (2023) show with idealised climate model simulations that the Northern Africa teleconnection mechanism plays the largest role in the Norwegian Earth System Model, while the ODT mechanism is absent. However, mechanisms might be different for other models, and it is important to keep in mind that widespread observations following major tropical eruptions are limited to the recent eruptions of Mount Agung, El Chichón and Mount Pinatubo, all coinciding with developing El Niño events, which might have conditioned the response Lehner et al. (2016).

Decadal prediction studies have found that the inclusion of volcanic forcing degrades the forecast skill in the Tropical Pacific region on multi-annual to decadal timescales (e.g. Timmreck et al., 2016; Meehl et al., 2015; Wu et al., 2023), indicating that models may not be realistically simulating part of the response to volcanic eruptions. Wu et al. (2023) show that following the eruptions of Agung, El Chichón and Pinatubo the observed tropical Pacific warming is better predicted by CESM1 decadal hindcasts that do not include the volcanic forcing, while decadal hindcasts (and non-initialised historical simulations) that include the volcanic forcing simulate a cooling that was not observed. Likewise, Schurer et al. (2023) show that in HadCM3 simulations the cooling response following major volcanic eruptions is overestimated unless the correct ENSO evolution is assimilated.

The Atlantic Ocean is another region of relevance following volcanic eruptions. The Atlantic multidecadal variability (AMV) is a North Atlantic basin-wide sea surface temperature (SST) fluctuation on decadal time scales (Knight et al., 2005). Previous





studies have shown that volcanic eruptions can impact the AMV via the direct surface cooling induced by the changes in radiation (Otterå et al., 2010; Swingedouw et al., 2017). The AMV can also respond indirectly to the volcanic eruptions in response to induced changes in the Atlantic Meridional Overturning Circulation (AMOC) and associated heat transports. Studies have shown that on multiannual to decadal timescales, the strength of the AMOC increases in response to large volcanic

eruptions (e.g. Stenchikov et al., 2009; Ding et al., 2014; Otto-Bliesner et al., 2016; Swingedouw et al., 2015; Hermanson et al., 2020; Fang et al., 2021). Two main mechanisms have been proposed to explain this strengthening: (1) an initial dynamical adjustment to the negative surface wind stress anomaly over the subpolar North Atlantic (Mignot et al., 2011; Zanchettin et al., 2012) and (2) reduced sea surface temperature and increased salinity (due to reduced precipitation) enhancing deep convection and a subsequent weakening of density stratification in the Labrador Sea (e.g. Iwi et al., 2012; Ortega et al., 2012; Stenchikov

et al., 2009). Despite the general consensus on the AMOC strengthening, its magnitude has been shown to be sensitive to the background conditions, eruption magnitude and climate model considered (Ding et al., 2014; Swingedouw et al., 2015).

This study will comprehensively analyse the climate response following the eruptions of Mount Agung (1963), El Chichón (1982) and Mount Pinatubo (1991) using a multi-model set of decadal predictions which follow a purposefully designed experimental protocol and builds upon the study of Hermanson et al. (2020). We analyse simulations from six decadal prediction

systems contributing to the CMIP6 Decadal Climate Prediction Project (DCPP Boer et al., 2016). The DCPP jointly with VolMIP (Zanchettin et al., 2016) defined a set of coordinated experiments (component C) directed toward understanding the influence and consequences of volcanic eruptions on decadal prediction. The fact that these simulations are decadal hindcasts which are initalised with the observed state, implies that the climate response might be more realistic (with respect to non-initialised simulations) and directly comparable to observations, as internal variability can modulate the response to the

volcanic forcing. With respect to the earlier analyses carried out in Hermanson et al. (2020), this study uses more recent models and a larger ensemble, which allows us to better detect potentially weak signals, and longer forecast outlooks up to nine years, which allows us to investigate the response on longer timescales. Another addition with respect to Hermanson et al. (2020) is that we explore the differences among the three eruptions and prediction systems, when possible. Finally, to fully exploit the decadal prediction protocol we compare the predicted surface temperature anomalies with observations to infer the importance

of including the volcanic forcing, attribute observed changes and determine to what extent the initial conditions can improve the agreement in the three hindcasts. The paper is organised as follows. Section 2 provides a description of the DCPP protocol, a description of the decadal forecast systems used and the statistical methods used. In section 3 we present the results evaluating the global mean and regional impacts of volcanic eruptions, focusing on the short and longer term responses and the particular responses in the Pacific and Atlantic oceans. The final section discusses and summarises the key results of this study.

## 2  Methods


In this study we use six state-of-the-art decadal predictions (the details of each system are summarised in Table 1) that follow the CMIP6 DCPP protocol A and C (Boer et al., 2016). The DCPP component A consists of 10-member ensembles of 10-year-long retrospective predictions initialised from an observation-based state every year from 1960 to 2018 that are forced with



| Model | Institution | Resolution (Atm and Oce) | Initialisation | Reference |
|---|---|---|---|---|
| CanESM5 | CCCma | T63L49 and 1°45L | Full-field | Sospedra-Alfonso et al. (2021) |
| CESM1-1-CAM5-CMIP5 | NCAR | 1°L30 and 1°60L | Full-field | Yeager et al. (2018) |
| CMCC-CM2-SR5 | CMCC | 0.9°×1.25°30L and 1°50L | Full-field | Nicolì et al. (2023) |
| EC-Earth3 | BSC | T255L91 and 1°75L | Full-field | Bilbao et al. (2021) |
| IPSL-CM6A-LR | IPSL | 2.5°x1.3°L79 and 1°75L | Anomaly | Estella-Perez et al. (2020) |
| HadGEM3-GC31-MM | MetOffice | N216L85 and 0.25°L75 | Full-field | Williams et al. (2018) |

**Table 1.** Details of the decadal prediction systems.

CMIP6 historical values of atmospheric composition and/or emissions. With the objective of isolating the impact of the major
volcanic eruptions that occurred during this period, the DCPP component C proposed to repeat the predictions initialised right
before the eruptions of Mount Agung (1963), El Chichón (1982) and Mount Pinatubo (1991) replacing the volcanic forcing
by the "background" volcanic aerosol forcing, computed as the mean over the period 1850-2014. This study analyses and
compares the two sets of prediction ensembles for the 1962, 1981 and 1990 start dates. This makes a total of 2 x 60 members to
characterise the multi-model response per eruption, and 2 x 180 members when the eruptions are combined. The impact of the
volcanic eruptions is determined by subtracting the predictions with and without the volcanic aerosols (DCPP-A - DCPP-C).
Since both prediction ensembles are initialised in the same way, we assume the forecast drift to be the same, and therefore
removed. These decadal prediction systems are initialised either the 30$^{th}$ of October or 31$^{st}$ of December, depending on the
model, therefore we discard the first two months of the those models initialised the 30$^{th}$ of October and consider January of the
year of the eruptions as the nominal start date.

These prediction systems use prescribed CMIP6 volcanic forcings (Thomason et al., 2018), except for CESM1-1-CAM5-
CMIP5, which uses prescribed volcanic forcing from Ammann et al. (2003). In these models the volcanic aerosols are repre-
sented by monthly-mean and zonal-mean aerosol optical properties, which are prescribed in the radiation code with a vertical
profile in the stratosphere. Figure 1 shows the CMIP6 stratospheric aerosol optical depth (AOD) at 530nm for the eruptions
of Agung in March 1963, for El Chichón in February 1982 and Pinatubo in June 1991. Note that while the main strato-
spheric AOD perturbation is due to these three large volcanic eruptions, the impact of smaller eruptions is also included in the
CMIP6 forcing. The global mean AOD (Figure 1a) shows that the 1991 eruption of mount Pinatubo was the largest eruption
of the three. The eruptions of Agung and El Chichón are more or less comparable in magnitude, although the tropical average
(30°N-30°S) for El Chichón is half the magnitude of the other two eruptions (Figure 1b). There are evident differences in the
geographical distribution of the forcing among the eruptions. The eruption of Pinatubo was approximately hemispherically
symmetric, while the eruption of Agung mainly affected the Southern Hemisphere and the eruption of El Chichón the Northern
Hemisphere (Figure 1 c and d). These differences are relevant to explain the climate impacts, as it will be discussed in later
sections.

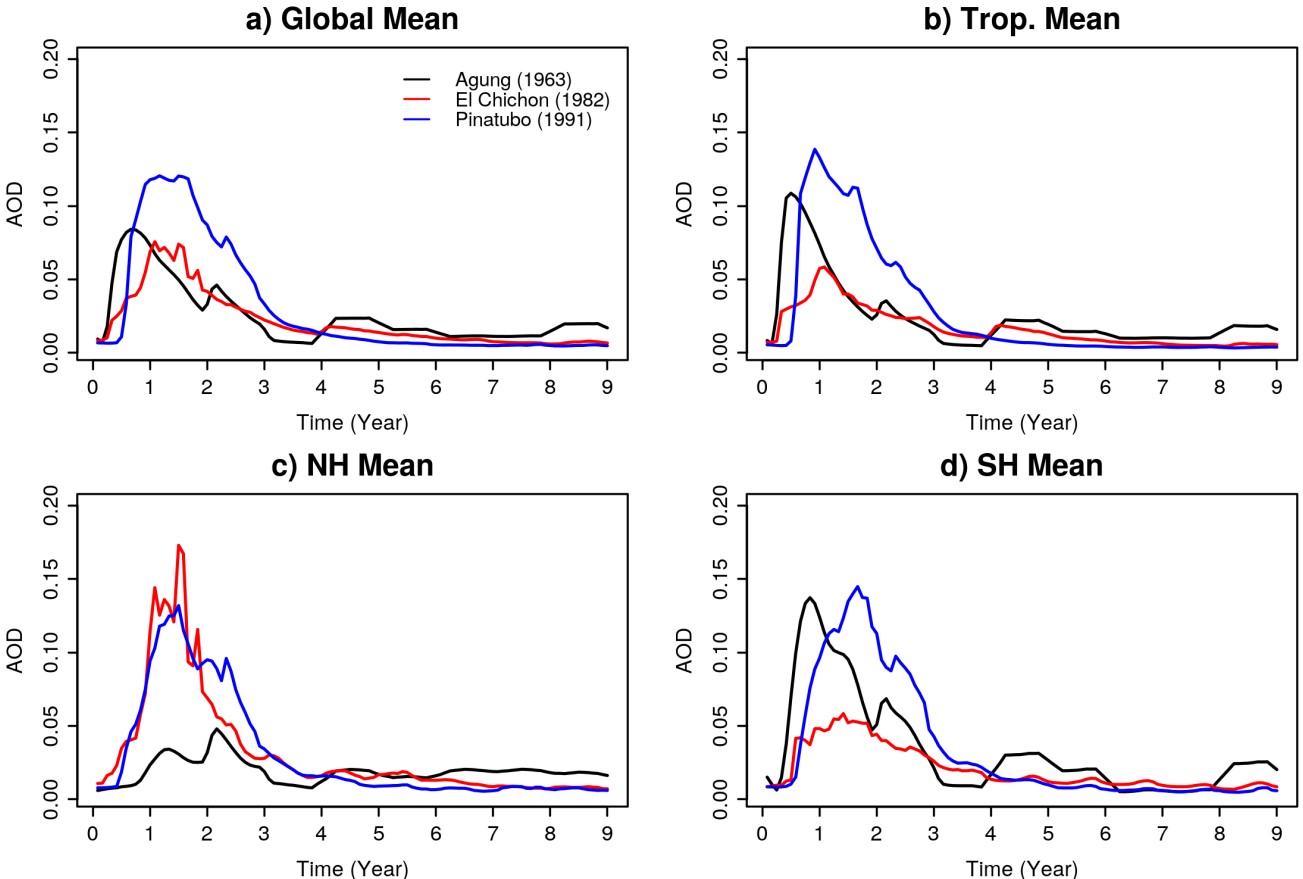

**Figure 1.** CMIP6 aerosol forcing optical depth at 530nm for the eruptions of Mount Agung, El Chichón and Mount Pinatubo, for different regions. t=0 is the eruption year for each eruption (starting in 1963, 1982 and 1991 respective).

In these models, as in the previous generation (Hermanson et al., 2020), ozone concentrations vary slowly and therefore the impacts arising from ozone depletion by the volcanic aerosols are not represented in these experiments.

The differences (DCPP-A minus DCPP-C), for both timeseries and fields, is significance tested by creating a distribution of 10-member mean differences by bootstrap with replacement of ensemble members from 1,000 repetitions. If the differences fall outside of the 2.5-97.5% quantile range of the distribution (equivalent to $p \leq 0.05$) we reject the null hypothesis (no difference between the DCPP-A and DCPP-C hindcasts) and the differences are considered significant. For the timeseries plots, the uncertainty is shown by the 10[th] and the 90[th] percentiles of the multi-model ensemble (formed by individual model members).

For the multi-model maps, in addition to the statistical significance of the differences, we also show the agreement among decadal prediction systems, which is shaded when all the models agree on the sign of the anomaly.

In this paper we compare the predicted surface temperature in the three hindcasts of DCPP-A and the DCPP-C experiments, initialised in 1963, 1982 and 1991, against observations. To compute the anomalies we compute the lead-time dependent



climatology for the period 1970-2005 using the DCPP-A decadal hindcasts initialised in 1960 to 2015. Forecast drift is assumed
to be equal in the DCPP-A and DCPP-C hindcast sets. The observational datasets used for verification are HadCRUT5 (Morice
et al., 2021) for near-surface temperature and HadSST.4.0.1.0 (Kennedy et al., 2019) for sea surface temperature. To determine
whether the predicted anomalies in the DCPP-A or the DCPP-C hindcast are closer to the observations we use the root mean
square error (RMSE). When considering spatial fields we compute the area-weighted RMSE. Note that here the RMSE is not
used as a forecast skill metric, but rather to compare the error in each of the hindcast sets and determine the impact of the
volcanic forcing.

## 3  Results

### 3.1  Global Mean Volcanic Response

Figure 2 shows the global mean net top-of-atmosphere (TOA) radiation flux response (DCPP-A minus DCPP-C) to the in-
dividual volcanic eruptions and the mean of the three. The TOA flux is computed as the incoming shortwave (i.e. the rsdt
CMIP6 variable) minus the outgoing shortwave (i.e. rsut) and longwave radiation (i.e. rlut) fluxes. The climate predictions
show a statistically significant post-volcanic decrease in global mean TOA which is consistent across the models, generally
peaking in the first boreal winter. There are differences however in the magnitude of the response and the recovery following
the three volcanic eruptions, coherent with the volcanic aerosol forcing (Figure 1). After the eruption of Agung, the negative
TOA anomalies reach approximately -1.7$\pm$0.4 W/m$^2$ (model-mean $\pm$ the inter-model standard deviation) and recover within
the next months. The TOA response for the eruption of El Chichón is the weakest and the only one in which the model spread
overlaps with the zero line, with the ensemble mean response reaching approximately -0.9$\pm$0.2 W/m$^2$ and recovering within
the next year. The response to the eruption of Pinatubo shows the strongest TOA negative anomalies of the three, reaching ap-
proximately -2.3$\pm$0.3 W/m$^2$ and recovering in approximately two years. These results are consistent with those of Zanchettin
et al. (2022).
In response to the negative TOA anomalies following the volcanic eruptions, the global mean surface temperature decreases
(Figure 3). The cooling anomalies peak in the second year, and subsequently recover in approximately 5 years. Similarly to
the TOA response, the prediction systems generally show a comparable global mean surface temperature response for the
individual eruptions, although noisier, and the differences are larger across the eruptions. For the eruptions of Agung and El
Chichón, the minimum global mean surface temperature anomalies are comparable (the model mean $\pm$ the inter-model standard
deviation is -0.21$\pm$0.05°C and -0.17$\pm$0.04°C respectively) even though the TOA anomalies differences are larger, indicating
there are potential non-linearities in the response or multiple mechanisms at play. The eruption of Mount Pinatubo shows
the largest global mean surface temperature anomaly, of -0.35$\pm$0.06°C (model mean $\pm$ the inter-model standard deviation),
coherent with the larger TOA anomalies. There is considerable inter-model spread in the response to both the Agung and El
Chichón eruptions, while for Pinatubo the models show stronger agreement, probably because of the greater signal-to-noise
ratio.

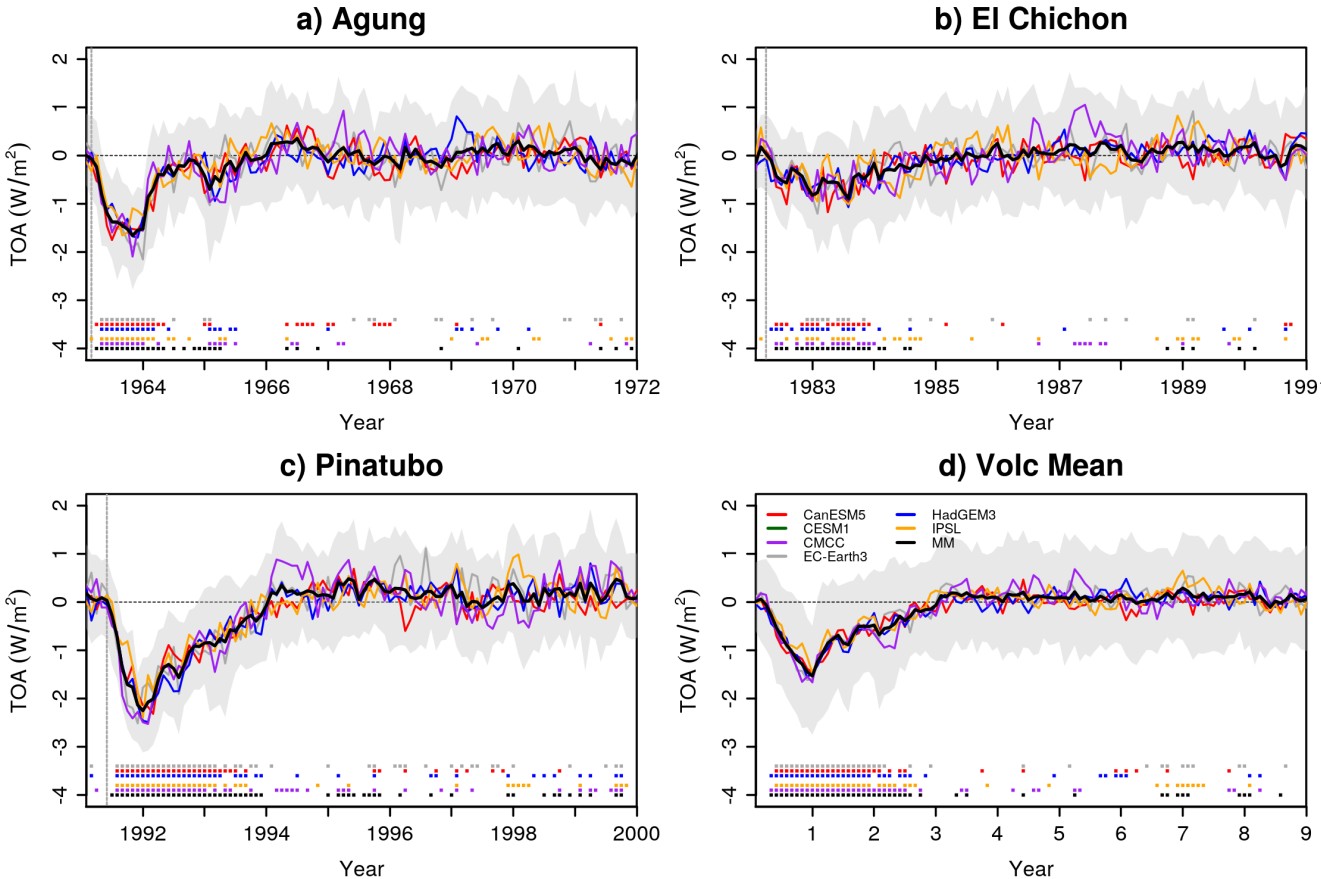

**Figure 2.** Global mean top-of-atmosphere radiation response (W/m$^2$) to the volcanic eruptions (DCPP-A minus DCPP-C). The ensemble mean for each model and the multi-model mean are shown. The shading is the multi-model member spread calculated as the 10[th] and 90[th] percentiles of the entire ensemble. Filled squares at the bottom part of the figure indicate statistically significant differences (see methods). The vertical grey dashed lines indicate the approximate time of the eruptions. The data for CESM1-1-CAM5-CMIP5 was not included since it is not available. Figure inspired by Zanchettin et al. (2022).

Figure 4 shows the changes in ocean heat content (OHC) in the upper 300 m, an integrated variable with large inertia for which the volcanic signals can persist longer in time Stenchikov et al. (e.g. 2009). The volcanic signals are detected throughout the whole decade following the eruptions, with minimum OHC values peaking around the fourth year following the eruption and a slow recovery that still yields statistically significant anomalies in the ensemble mean by the end of the forecasts in each

of the individual eruptions. Likewise, all systems simulate long-lasting significant responses, with some inter-model differences in the peak timing and the recovery rate.

Having characterised the global mean response to the three volcanic eruptions, we compare the global mean near-surface air temperature anomalies predicted in both the DCPP-A and DCPP-C hindcasts against observations. Figure 5 shows that overall

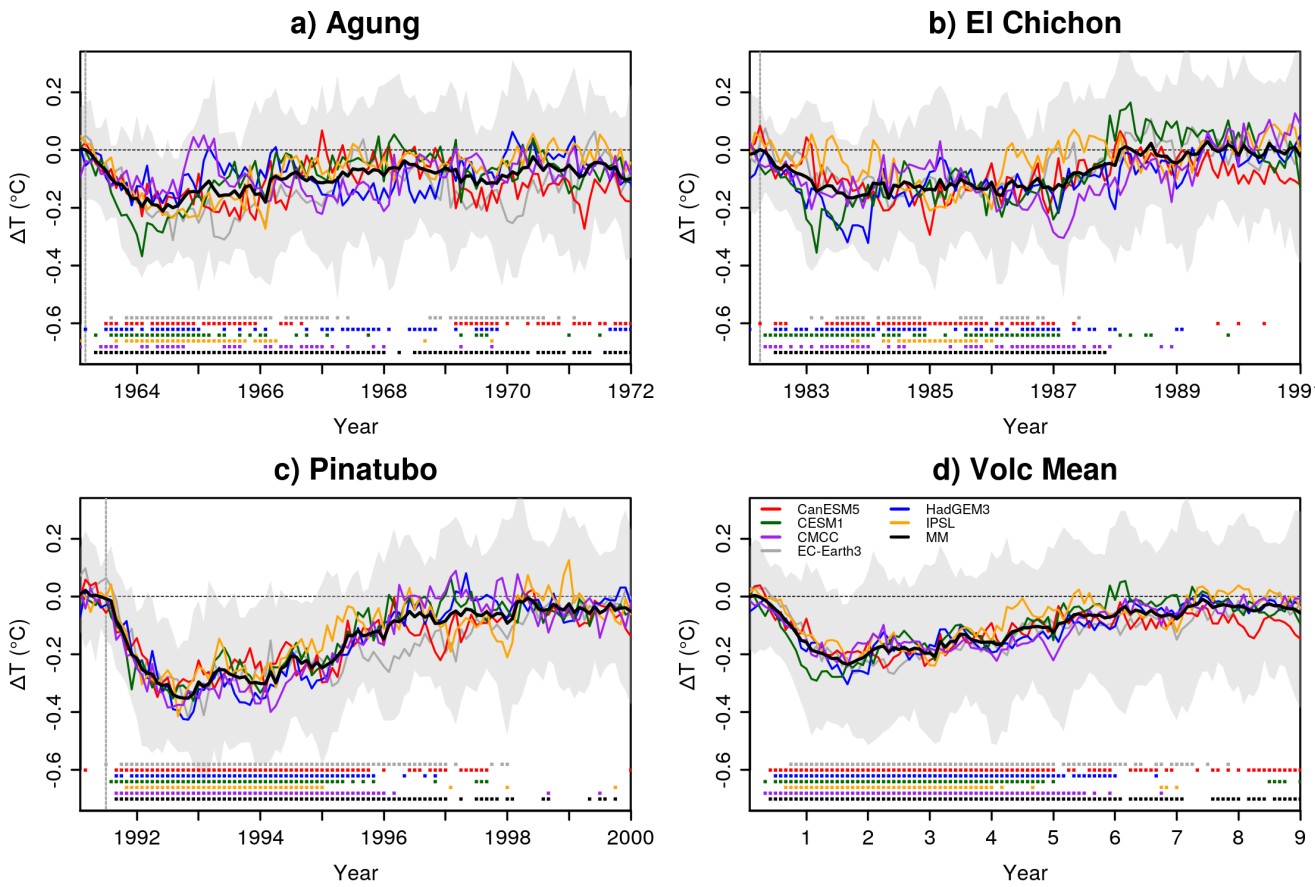

**Figure 3.** Global mean surface air temperature response (°C) to the volcanic eruptions (DCPP-A minus DCPP-C). The ensemble mean for each model and the multi-model mean are shown. The shading is the multi-model member spread calculated as the 10$^{th}$ and 90$^{th}$ percentiles of the entire ensemble. Filled squares on the bottom part of the figure indicate statistically significant differences (see methods). The vertical grey dashed lines indicate the approximate time of the eruptions.

the DCPP-A predictions, which include the volcanic forcing, reproduce the HadCRUT5 temperature anomalies more closely
than the DCPP-C. The DCPP-A hindcasts generally simulate the observed cooling tendencies in the initial years following
the eruptions and the observed anomalies (from HadCRUT5) are generally within the 10$^{th}$ and 90$^{th}$ percentiles. In contrast,
the DCPP-C hindcasts are generally warmer and observations tend to fall more frequently below the 10$^{th}$ and 90$^{th}$ percentiles.
This is shown by lower RMSE of the DCPP-A hindcasts in comparison to the DCPP-C (Table S1). In particular, the volcanic
forcing following the eruption of Pinatubo is remarkably important to simulate the observed global mean surface temperature
variability in the early 90s, as without the forcing the observations are out of the models' envelope the two years following the
eruption.

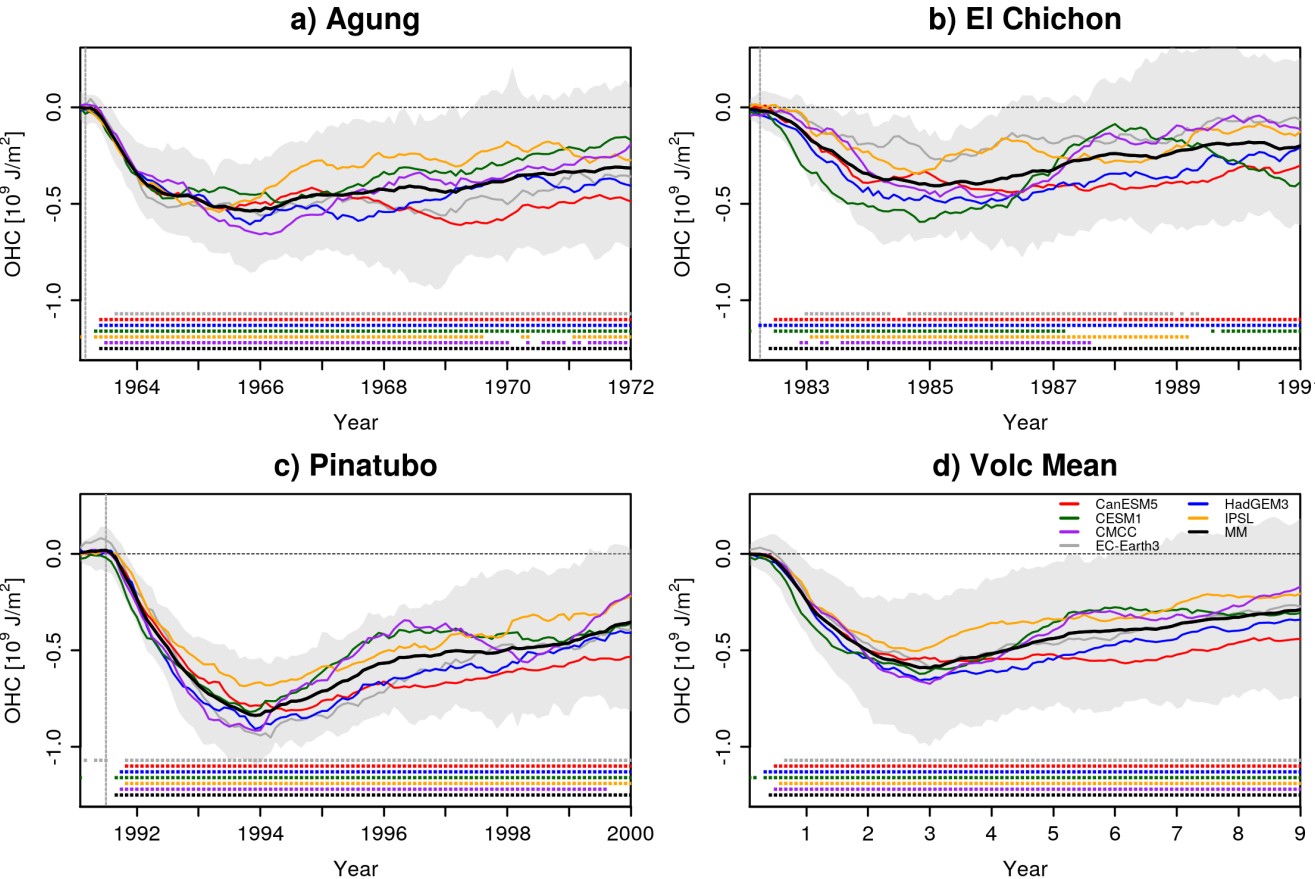

**Figure 4.** Global ocean heat content response (J/m$^2$) to the volcanic eruptions (DCPP-A minus DCPP-C). The ensemble mean for each model and the multi-model mean are shown. The shading is the multi-model member spread calculated as the 10$^{th}$ and 90$^{th}$ percentiles of entire ensemble. Filled squares at the bottom part of the figure indicate statistically significant differences (see methods). The vertical grey dashed lines indicate the approximate time of the eruptions.

## 3.2 Spatiotemporal Characteristics of the Volcanic Response

The surface air temperature response in the first year (computed June-May to characterise the post-eruption anomalies) following the volcanic eruptions (Figure 6) shows a distinct pattern, largely consistent across the volcanic eruptions (as shown by the shading in Figure 6). It is characterised by a generalised cooling, largest in the Tropics and subtropics, and a warming in the Eurasian Arctic sector (discussed further in section 3.3). The magnitudes of the anomalies tend to be greater over land than over the ocean, in agreement with response to radiative forcing (e.g. Eyring et al., 2021). Despite the overall similarities in the surface temperature response among the eruptions, there are some differences which can be explained by the magnitude and geographical pattern of the TOA anomalies following each eruption (Figure S1). For the eruption of Agung the TOA anomalies mainly occur in the Southern hemisphere (Figure S1a), and therefore the surface temperature anomalies are larger over

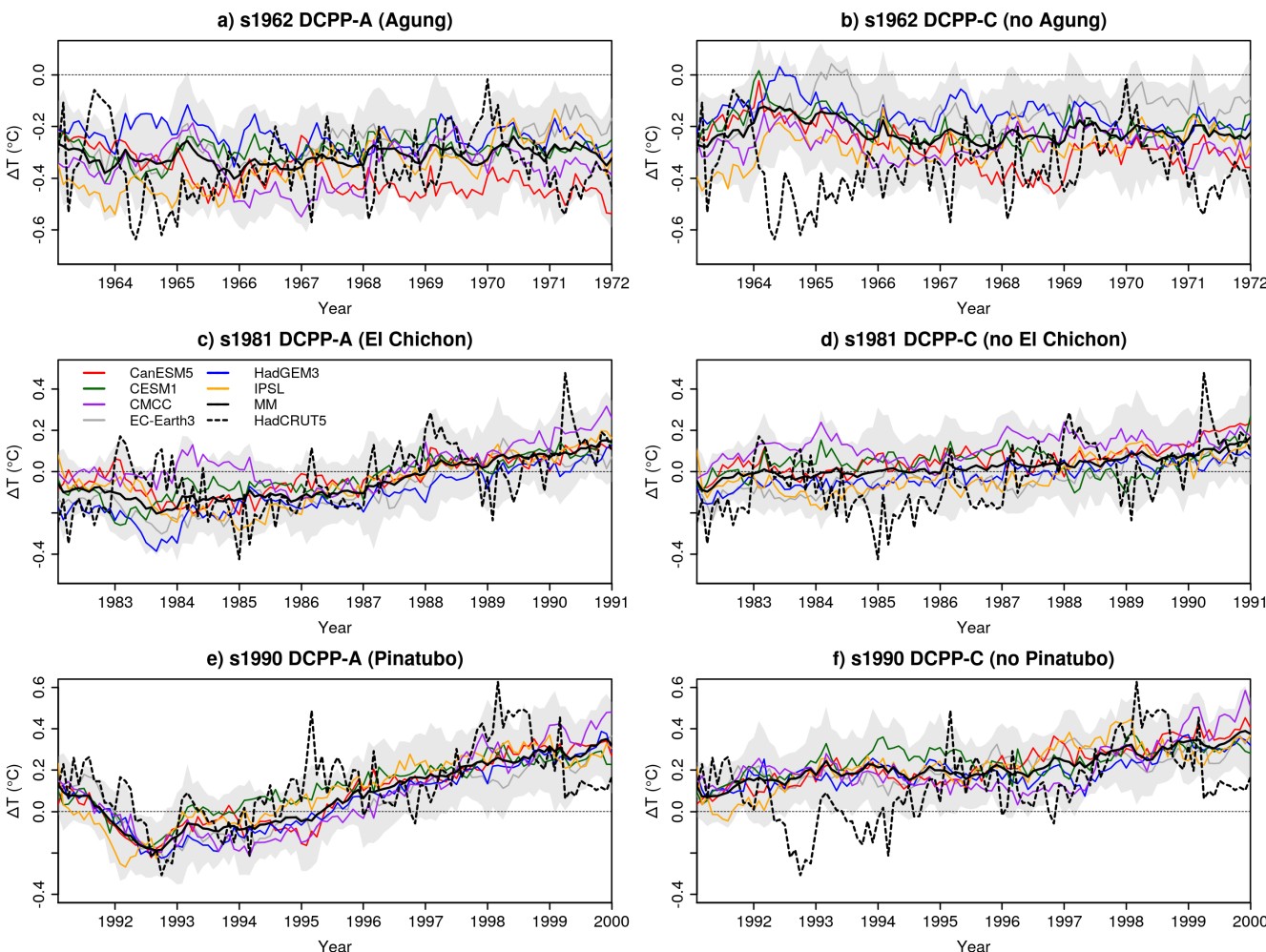

**Figure 5.** Monthly mean global near-surface temperature anomalies (°C) of the predictions initialised in 1962, 1981 and 1990 for the DCPP-A (with volcanic forcing) and DCPP-C (without volcanic forcing) experiments. HadCRUT5 is used as the observational reference (dashed line). The anomalies have been computed with respect to the period 1970-2005 (see methods for further information). The shading is the multi-model member spread calculated as the 10th and 90th percentiles of the entire ensemble.

the tropics and Southern hemisphere (Figure 6a). In contrast, for the eruption of El Chichón the TOA anomalies occur mainly in the Northern hemisphere (Figure S1b) and the surface temperature anomalies are larger over the tropics and the Northern hemisphere (Figure 6b). The eruption of Pinatubo caused a hemispherically symmetric response (Figure S1c) with surface temperature anomalies that were stronger over the tropics and at high latitudes (Figure 6c).

For years 2-5 (computed as June-May), the cooling spreads worldwide in response to the volcanic forcing (Figure 7a-d). There are evident differences among the eruptions, associated with the magnitude and geographical pattern of the forcing as described previously. For the eruption of Agung the largest surface temperature anomalies are located over the tropics and







**Figure 6.** Model mean near-surface air temperature (°C) response (DCPP-A minus DCPP-C) during the first year following the volcanic eruptions (June-May). Hatching indicates statistically significant anomalies, while the shading indicates model agreement (see methods).

especially in the Southern Hemisphere sector (Figure 7a), while following the eruption of El Chichón the strongest surface temperature anomalies are located over the polar latitudes of the Northern Hemisphere probably linked to the Arctic amplifica-
tion phenomenon (Figure 7b). The eruption of Pinatubo shows stronger surface temperature anomalies worldwide, albeit with maxima over the Arctic like for El Chichón (Figure 7c). The mean response to the three eruptions is consistent with the one to Pinatubo, with surface temperature anomalies that are statistically significant worldwide (Figure 7d).

At later forecast times (years 6-9) the surface temperature anomalies largely decrease in magnitude (Figure 7e-h), consistent with the recovery time-scale after the eruptions (Figure 3). The response on these time-scales is partly related to the magnitude





of the eruption. For the eruption of Agung we find that negative surface temperature anomalies persist over the tropics and
the Southern Hemisphere (Figure 7e). For the eruption of El Chichón, which is the weakest of the three, there are barely
any significant anomalies remaining (Figure 7f). Finally, for the eruption of Pinatubo, we find some regions with significant
anomalies which are particularly strong over the Arctic (Figure 7g).

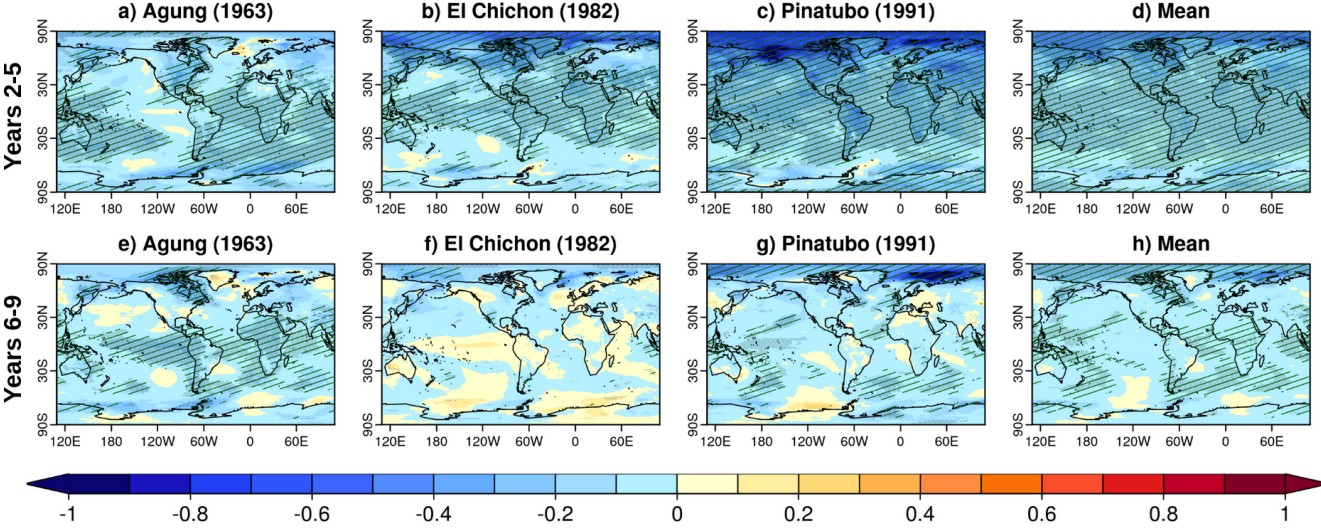

**Figure 7.** Multi-model mean near-surface air temperature (°C) response (DCPP-A minus DCPP-C) during years 2-5 (first row) and years
6-9 (second row) following the eruptions of Mount Agung (1963), El Chichón (1982), Mount Pinatubo (1991) and the mean of the three
volcanoes (left to right). The annual mean is defined from June to May. Hatching indicates statistically significant anomalies, while shading
indicates model sign consistency (see methods).

To determine the importance of volcanic forcing at the regional scale, we evaluate the predicted surface temperature anoma-
lies in the three DCPP-A and DCPP-C hindcast sets against the HadCRUT5 observational dataset. We focus on the multi-model
ensemble mean for year 1, years 2-5 and years 6-9. Overall, we find that the volcanic forcing only leads to a generalised im-
provement in forecast years 2-5, as indicated by the lower global RMSE the three DCPP-A hindcasts with respect to DCPP-C
(Table S2; by 12% for Agung and El Chichón and 16% for Pinatubo). Figure 8 shows that the multi-model mean surface tem-
perature anomaly patterns for years 2-5 (when the volcanic radiative impact is strongest) predicted by the DCPP-A hindcasts
are cooler than the DCPP-C, as expected, and closer to the observations. In comparison with the observed anomaly patterns, the
multi-model predicted pattern is smoother and does not reproduce most of the regional variations. This is probably because at
this forecast range the multi-model pattern is mostly due to the radiatively forced response and the observed regional variations
are due to noise. For the first forecast year and forecast years 6-9, although the volcanic forcing has a distinct surface temper-
ature imprint following the eruptions (Figures 6 and 7), the fact that we do not find a detectable improvement in the DCPP-A
hindcasts over DCPP-C (Table S2) might be because: (1) the forecast error at the regional level is greater than the volcanic





impact, (2) the local volcanic response is overwhelmed by internally generated variability and/or (3) the regional response to the volcanic forcing is not correctly simulated by the models.

**Figure 8.** Near-surface air temperature anomalies (°C) for years 2 to 5 of the multi-model mean predictions initialised in 1962, 1981 and 1990 for the DCPP-A (with volcanic forcing) and DCPP-C (without volcanic forcing) experiments and HadCRUT5. The anomalies have been computed with respect to the period 1970-2005 (see methods). Note that the anomalies are computed from June-May.

### 3.3 Northern Hemisphere Atmospheric Response

The models consistently simulate a surface warming over North Eurasia and/or the Barents-Kara seas in the first year after the eruptions (Figure 6), although it is only statistically significant for Agung and the composite of the three eruptions. A focus on the seasonal surface temperature changes reveals that after the eruptions of Agung and Pinatubo, the surface warming





over North Eurasia and the Barents-Kara seas reaches its maximum magnitude and extent in the first winter (Figure 9a and

c), although only small regions show statistically significant anomalies. Interestingly, for the Agung eruption the warming is
already present in the first JJA and SON (Figure S2a-d). In contrast, following the eruption of El Chichón there are no warm
anomalies present over the Eurasian continent in the first DJF (Figure 9b), which shows cool anomalies instead (not statistically
significant) and very weak and localised warm anomalies over the Barents-Kara seas. Due to the disparate responses among
the eruptions, the composite of the three eruptions shows positive yet not significant anomalies both in DJF and the earlier
seasons (Figure S2). These results are consistent with studies suggesting that the Eurasian warming might be too weak and
hence overwhelmed by unforced variability to all major eruptions (DallaSanta and Polvani, 2022).

Coherent with the surface temperature anomalies, for the Agung and the Pinatubo eruptions the sea level pressure anomalies
feature a positive NAM-like pattern during the first post-eruption winter (Figure 9e and g), not occurring for the eruption of El
Chichón (Figure 9f). The positive NAM-like pattern in the first post-eruption winter is associated with the warming of the trop-
ical lower stratosphere in the months following the volcanic eruptions (Figure 10a-d) which increases the poleward temperature
gradient and might explain the acceleration of the polar vortex (Figure 10e-h), in line with previous studies (e.g. Hermanson
et al., 2020). The lower stratospheric temperatures in the tropics show a strong increase following the three eruptions, peaking
approximately six months after the eruption and lasting for three years (Figure 10a-d), thereby increasing the poleward tempera-
ture gradient (not shown). Four of the forecast systems (CanESM5, CMCC-CM2-SR5, EC-Earth3 and HadGEM3-GC31-MM)
cluster together, while IPSL-CM6A-LR and CESM1-1-CAM5-CMIP5 tend to simulate greater anomalies in response to the
eruptions. In the case of CESM1-1-CAM5-CMIP5 this could be because this model was forced with CESM-specific volcanic
forcing (Ammann et al., 2003), rather than CMIP6. The temperature anomalies are coherent with the magnitude of the volcanic
forcings; the multi-model mean reaches $4.7\pm0.7°C$, $2.4\pm1.4°C$ and $5.6\pm1.7°C$ for the eruptions of Agung, El Chichón and
Pinatubo respectively.

Accompanying the stratospheric warming there is a detectable acceleration of the polar vortex following the eruptions of
Agung and Pinatubo in the first winter in both the model mean and the eruption mean (Figure 10e, g and h). Not all individual
models show this acceleration though, probably because a larger ensemble size is needed to overcome the low signal-to-
noise ratio and detect the response. Furthermore, a stronger stratospheric temperature response does not necessarily produce a
stronger polar vortex response. In the case of the eruption of El Chichón the warm anomalies in the tropical lower stratosphere
are weaker (Figure 10b), which does not seem to result in an acceleration of the polar vortex and explains why a positive phase
of the NAM with its associated warming over Eurasia are not simulated (Figure 10f). The sea level pressure anomalies do
resemble a positive NAM-like pattern in Autumn, when weak warm anomalies are present over North Eurasia (Figures S2f and
S3f), but this is not accompanied by a detectable acceleration of the polar vortex.

### 3.4 Response in the Pacific Ocean

Many studies have reported the impact of volcanic eruptions on the variability of the Pacific Ocean from seasonal-to-decadal
timescales. We start by documenting the impact of volcanic eruptions on ENSO in our simulations. To isolate the dynamical
response of ENSO from the surface cooling effect we define the Niño 3.4 SST index relative to the tropical SST mean (20N-

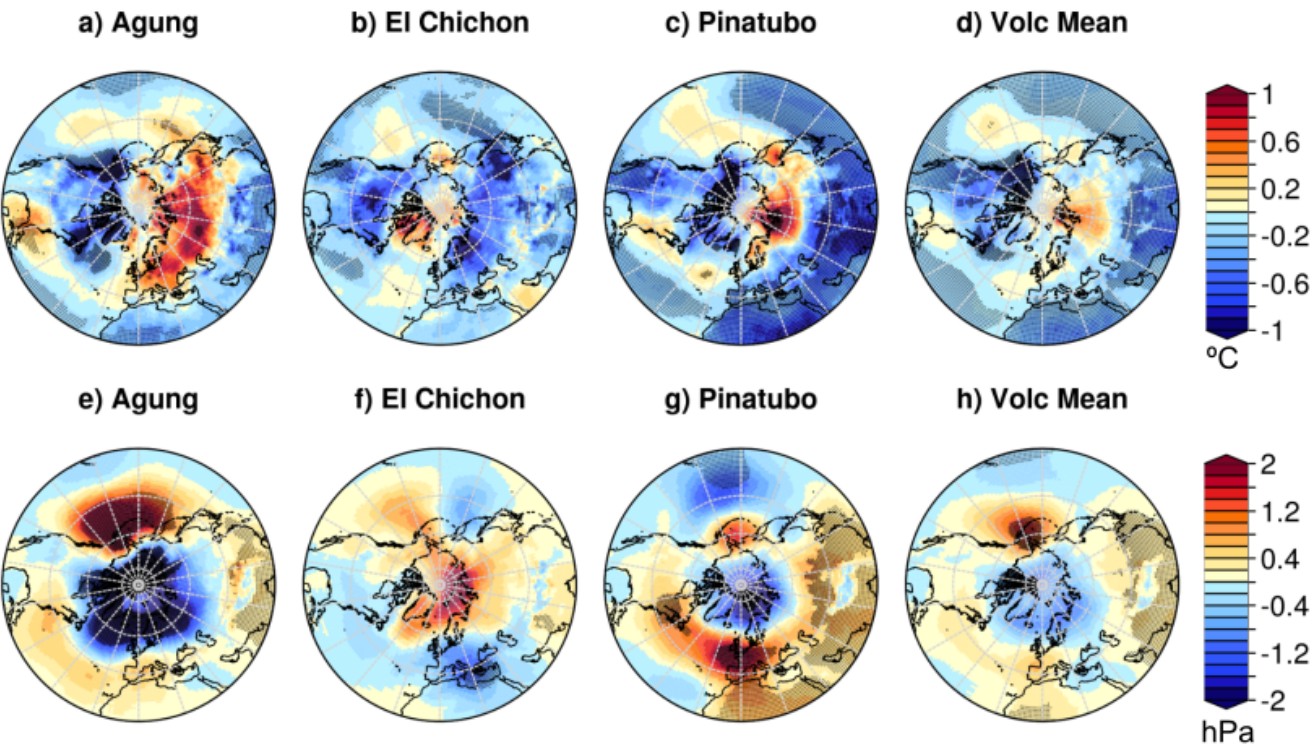

**Figure 9.** Multi-model and multi-eruption response (DCPP-A minus DCPP-C) of surface air temperature (a-d) and sea level pressure (e-h) in the first boreal winter (DJF) following the volcanic eruptions. The ensemble mean for each model and the multi-model mean are shown. Shading indicates statistically significant anomalies.

20S) as in Khodri et al. (2017). Figure 11 shows that there is a large uncertainty in the relative Niño3.4 response to the volcanic eruptions and no consistent response across the individual models. This is partly due to the small ensemble size for the individual models (10 ensemble members), since large ensembles have been previously shown to be required to detect such signals (e.g. Ménégoz et al., 2018; Hermanson et al., 2020). Only when considering the model-mean and multi-eruption
composite we find a clear and statistically significant response. It is characterised by the development of weak El Niño-like conditions in the year of the eruptions which peak in the following year and then transitions to La Niña-like conditions in the second and third years following the eruption. There are, however, some differences among the volcanic eruptions, as the multi-model El Niño response is stronger and significant only for the eruptions of Agung and Pinatubo (Figure 11a and c), while the delayed La Niña response is clear and significant for Pinatubo and marginally significant (i.e. only during a season) for El
Chichón (Figure 11b). From the individual models CMCC-CM2-SR5 is the only one consistently showing the El Niño-like and La Niña-like responses after the three eruptions; all the rest are not significant in almost all cases. HadGEM3-GC31-MM and CESM1-1-CAM5-CMIP5 also show a significant positive ENSO response in the multi-eruption composite, with CESM1-





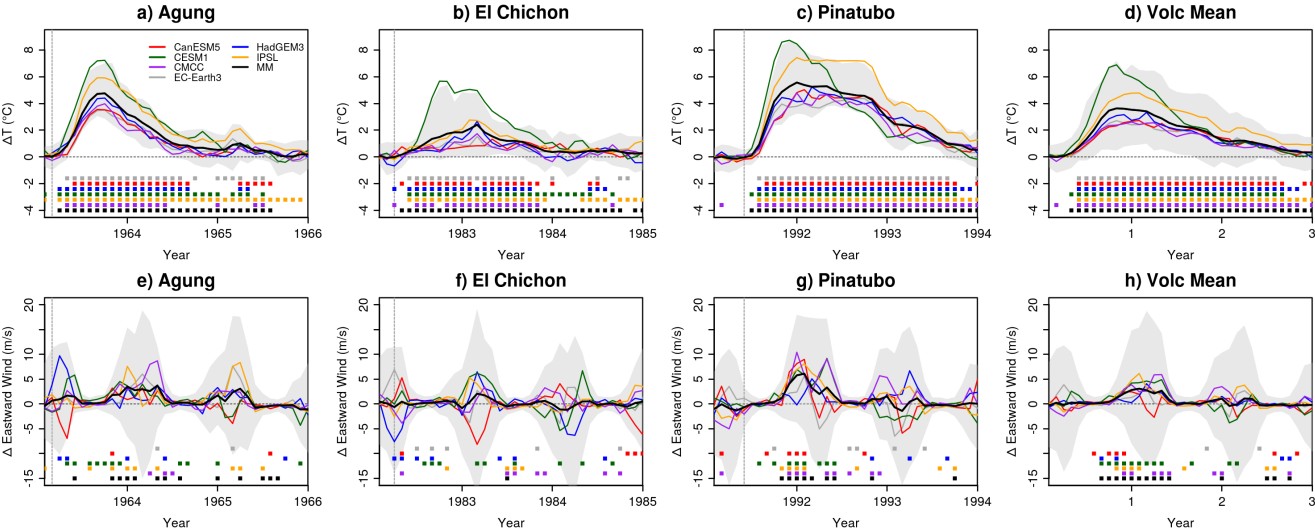

**Figure 10.** Stratospheric air temperature in the tropics (30°N - 30°S at 50 hPa) and polar vortex (average zonal velocity over 55°N–75°N at 50 hPa) response (DCPP-A minus DCPP-C) following the volcanic eruptions. The ensemble mean for each model and the multi-model mean are shown. The shading is the multi-model member spread calculated as the 10$^{th}$ and 90$^{th}$ percentiles. Filled squares on the bottom part of the figure indicate statistically significant differences (see methods).

1-CAM5-CMIP5 showing a subsequent significant La Niña-like response. None of the other three systems simulate the El Niño/La Niña response.

To further explore the development of the El Niño-like conditions we look at the model-mean composite responses in surface air temperature and precipitation in the tropics in JJA and DJF of the first and second years following the eruptions. The eruption-mean response shows the development of a Niño-like state from the first JJA to the second DJF (Figure 12), comparable to the results shown in Hermanson et al. (2020). In the year of the eruption, the tropics cool, especially over the continents, and there is a reduction in precipitation over Africa and the Maritime Continent (Figure 12 b and d). In the following

year's winter, the East and Central Equatorial Pacific show enhanced warming relative to the rest of the tropics, accompanied by the characteristic El Niño-like precipitation pattern (Figure 12 g and h).

While combining the different eruptions is a way to increase the ensemble size and improve the detection of weak signals, in this case it is probable that due to the different characteristics of each eruption, timing and background climate state, the mechanisms at play are not the same for each eruption and therefore their impact on ENSO also changes. It is beyond the scope

of this study to determine which mechanisms might dominate the response in the individual eruptions since these seem to vary from one model to another (Figure 11a-c). Nonetheless we find that overall, for the eruptions of Agung and Pinatubo, the El Niño-like state develops and peaks in the first year following the eruptions, while for the eruption of El Chichón the El Niño-like state occurs in the same year of the eruption. In the case of the Agung, initially a weak La Niña-like conditions develop in the year of the eruption (Figure S4a and b) accompanied by a northern shift of the ITCZ (Figure S5a and b), then the conditions

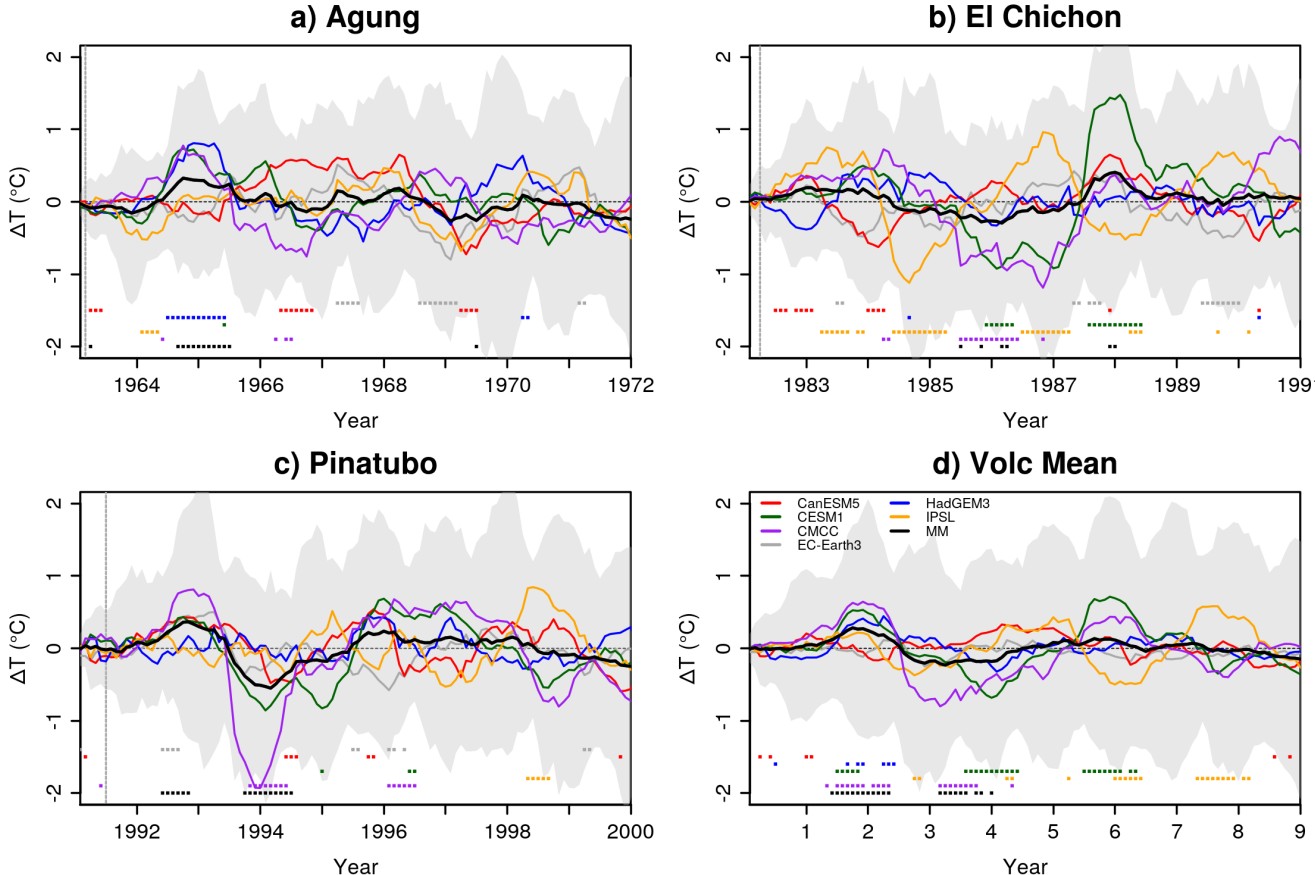

**Figure 11.** Relative Niño3.4 index response following the eruptions of a) Mount Agung (1963), b) El Chichón (1982), c) Mount Pinatubo (1991) and d) the mean of the three eruptions. Filled squares on the bottom part of the figure indicate statistically significant differences (see methods). The ensemble mean for each model and the multi-model mean are shown. The shading is the multi-model member spread calculated as the 10th and 90th percentiles. The vertical dashed lines indicate the approximate time of the eruptions.

shift to El Niño-like with a lack of cooling relative to the rest of the tropics (Figure S4c and d) and accompanied by increased precipitation (Figure S5c and d). For the eruption of El Chichón, we find an El Niño-like state developing and peaking in the year of the eruption (Figure S4e and f). The responses to the eruptions of Agung and El Chichón are broadly consistent with the results of Pausata et al. (2020), who show using idealised simulations with NorESM1-M that the ENSO response to NH and SH eruptions differs due to shifts in the Inter-tropical Convergence Zone (ITCZ) as a result of the asymmetric hemispheric

cooling. Finally, for the eruption of Pinatubo, which due to its intensity and location had an approximate hemispherically symmetric forcing and also induced the strongest cooling anomalies (Figure S4j-m), El Niño-like conditions develop and peak in the year following the eruption. In this case the response is in broad agreement with the mechanism proposed in Khodri et al.





(2017), which starts with reduced precipitation over tropical Africa (Figure S5j and k) leading to the propagation of anomalous atmospheric Kelvin waves that weaken the trade winds over the western Pacific.

The ocean state has been found to be another relevant factor modulating the ENSO response in climate model simulations (e.g. Predybaylo et al., 2020). Observations show that in the months prior to the three volcanic eruptions considered in this study, El Niño phases were already developing, peaking later in the first winter following the eruptions (e.g. McGregor et al., 2020). Figure S6 shows that the observed ENSO anomalies in the first months after the eruptions were reasonably well predicted for Agung, less so for El Chichón, but not for Pinatubo, in both DCPP-A and DCPP-C sets. The very small differences

identified between both forecast ensembles suggest that the volcanic forcing has a weak impact, which could imply either that the observed ENSO signal in the first year was not forced by the eruptions, or that the models systematically fail to simulate the true mechanism of response. At later forecast times, the models also misrepresent the observed ENSO anomalies.

     Previous studies have shown that including the volcanic forcing in decadal climate predictions can degrade the skill of SST over the central-eastern tropical Pacific Ocean on multiyear-to-decadal timescales (Timmreck et al., 2016; Wu et al., 2023).

In these predictions, the Tropical Pacific SST (averaged over 20°S–20°N, 120°E–80°W as in Wu et al. (2023)) experiences a net cooling during the first four years following the eruptions, although with important differences among the models and volcanic eruptions regarding the magnitude of the cooling (Figure S7). At later forecast times (years 6-10) there is no significant response to the volcanic forcing, with no evident differences between the ensembles. To evaluate whether the volcanic forcing has a detrimental impact on these predictions we compare the forecast anomalies of the DCPP-A and DCPP-C hindcasts with

SST observations.Figure 13 shows that for the eruptions of Agung and El Chichón the differences between the DCPP-A and DCPP-C hindcasts is small (Table S3), although including the volcanic aerosols may improve the initial years of the forecasts. For the eruption of Agung, 1964 is too warm unless volcanic aerosols are included (Figure 13a and b), and for the eruption of El Chichón, the multi-model tendency for 1983-1985 is better reproduced in DCPP-A compared to DCPP-C (Figure 13c and d). In contrast, for the eruption of Pinatubo, including the volcanic forcing negatively impacts the prediction initialised in 1990

in all systems by causing a local cooling that was not observed. This is also evident from the RMSE values shown in table S3.

### 3.5    Response in the North Atlantic Ocean

The North Atlantic Ocean is a region where the impact of volcanic eruptions has been shown to persist on decadal timescales (e.g. Ortega et al., 2012; Swingedouw et al., 2017; Hermanson et al., 2020; Borchert et al., 2021; Fang et al., 2021). Volcanic eruptions will always tend to cause a direct cooling of North Atlantic SSTs via the worldwide reduction in incoming shortwave

radiative fluxes (e.g. Swingedouw et al., 2017). As shown for the global mean surface temperature, in the North Atlantic Ocean, the SST decreases following the eruptions with a subsequent recovery (Figure S8). There are important differences regarding the magnitude of the cooling and the recovery time, which are larger across the volcanic eruptions than across models. Comparing the predicted SST anomalies of the DCPP-A and DCPP-C hindcasts with SST observations shows that overall the volcanic forcing positively impacts the predictions (also shown by the RMSE values in table S4). This is particularly relevant

following the eruption of Pinatubo, for which including the volcanic forcing is necessary to realistically simulate the North Atlantic SST variability in all models, at least in the first few years (Figure 14).







**Figure 12.** Multi-model and multi-eruption surface air temperature (°C) and precipitation (mm/day) responses following the eruptions in the tropics. Hatching indicates statistically significant anomalies (see methods).





**Figure 13.** Tropical Pacific (20°S–20°N, 160°E–80°W) SST anomalies (°C) in the predictions initialised in 1962, 1981 and 1990 for the DCPP-A (with volcanic forcing) and DCPP-C (without volcanic forcing) experiments. HadISSTv4 is used as the observational reference (dashed line). The anomalies have been computed with respect to the period 1970-2005 (see methods for further information). The ensemble mean for each model and the multi-model mean are shown. The shading is the multi-model member spread calculated as the 10th and 90th percentiles of the entire ensemble.

To isolate the changes that are specific to the North Atlantic Ocean, which might additionally arise from internal variability processes, we compute the AMV index as a standardised anomaly relative to the global SST mean (between 60°N-60°S) (Trenberth and Shea, 2006). We find that for this AMV index there is no significant impact due to the volcanic forcing (Figure S9). As previous studies have shown, the AMOC plays a key role modulating the AMV, associated with changes in ocean transport convergence on timescales longer than ten years (e.g. Knight et al., 2005; Kim et al., 2018; Oelsmann et al., 2020; Fang et al., 2021). Therefore, the predictions might be too short to reflect an impact due to the volcanic forcing on the AMV

**Figure 14.** North Atlantic (0°N-60°N, 80°W-0°) SST anomalies (°C) in the predictions initialised in 1962, 1981 and 1990 for the DCPP-A (with volcanic forcing) and DCPP-C (without volcanic forcing) experiments. HadISSTv4 is used as the observational reference (dashed line). The anomalies have been computed with respect to the period 1970-2005 (see methods for further information). The ensemble mean for each model and the multi-model mean are shown. The shading is the multi-model member spread calculated as the 10[th] and 90[th] percentiles of all members.

(e.g. Swingedouw et al., 2015; Otto-Bliesner et al., 2016; Borchert et al., 2021). To further explore if the AMV response could be explained by changes in North Atlantic Ocean dynamics, we investigate the changes in the mixed layer depth and the AMOC 360 (i.e. precursors to AMV changes). First we consider the multi-model and multi-eruption composites for the mixed layer depth in the subpolar North Atlantic. Figure 15a-d shows a significant enhancement of the deep mixed layer in late winter and early spring (when the mixed layer depth in this region attains its maximum) in the first three years following the eruptions, with a maximum in the Labrador Sea (Figure 15a-d). The mixed layer depth is usually interpreted as a proxy for deep convection,



and in many models enhanced deep mixing in the subpolar North Atlantic is associated with strengthening of the AMOC a few
years later (e.g. Dong and Sutton, 2005; Ortega et al., 2011). Indeed, Figure 15e-h shows that the mean AMOC streamfunction
also experiences a small but significant strengthening in response to the volcanic forcing during years 2-9 after the eruption.
The sign of the response is consistent for all the individual eruptions and the magnitude of both the mixed layer deepening
and the AMOC strengthening seems to depend on the magnitude of the volcanic forcing, with responses in order of decreasing
magnitude for Pinatubo, Agung and El Chichón.

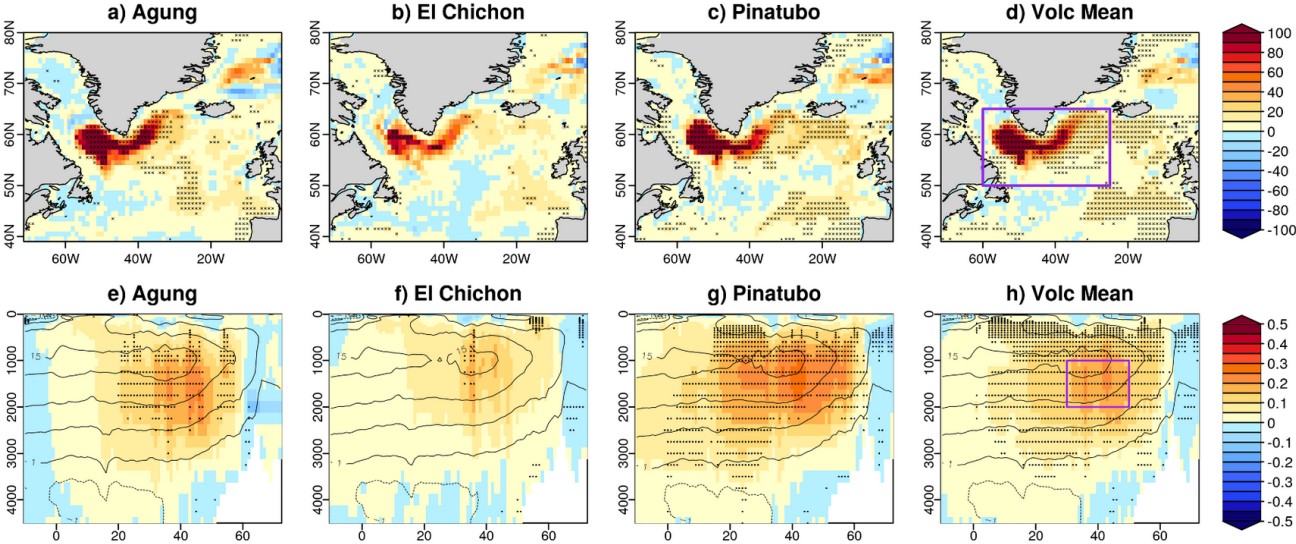

**Figure 15.** Multi-model and multi-eruption composites response for the mixed layer (February-March-April) depth (m) for years 1-3 and
overturning stream function (Sv) years 2-9 to the volcanic eruptions. Stippling indicates statistically significant anomalies (see methods).

We now explore the temporal evolution of the responses in the whole ensemble for two indices of the mixed layer depth and
AMOC by averaging over the purple boxes in Figure 15d and h respectively. Figure 16 shows that the response of these indices
has a large spread among the models, with HadGEM3-GC31-MM and CMCC-CM2-SR5 dominating the multi-model signal
and the rest of the models not simulating a consistent nor a significant response. Hence, Figure 15 should be interpreted with
care as the result is dominated by these models.

To further explore the origin of the inter-model differences in the AMOC response we look at the background density
stratification conditions in each model to infer their preconditioning role on convection. Figure 17a-c shows the change in the
multi-eruption composite profiles of temperature, salinity and density in the Labrador Sea due to the effect of the volcanic
eruptions, in the first three winters (DJF) after their occurrence. We focus on DJF, which is a couple of months before the peak
season for convection, because it includes both the preconditioning signal and the response to the vertical mixing, which hinders
the interpretation of the results. CMCC-CM2-SR5, HadGEM3-GC31-MM and IPSL-CM6A-LR show significant changes in
temperature, salinity and density in response to the volcanic forcing, which mostly imply a cooling, a salinification and a

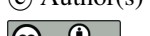



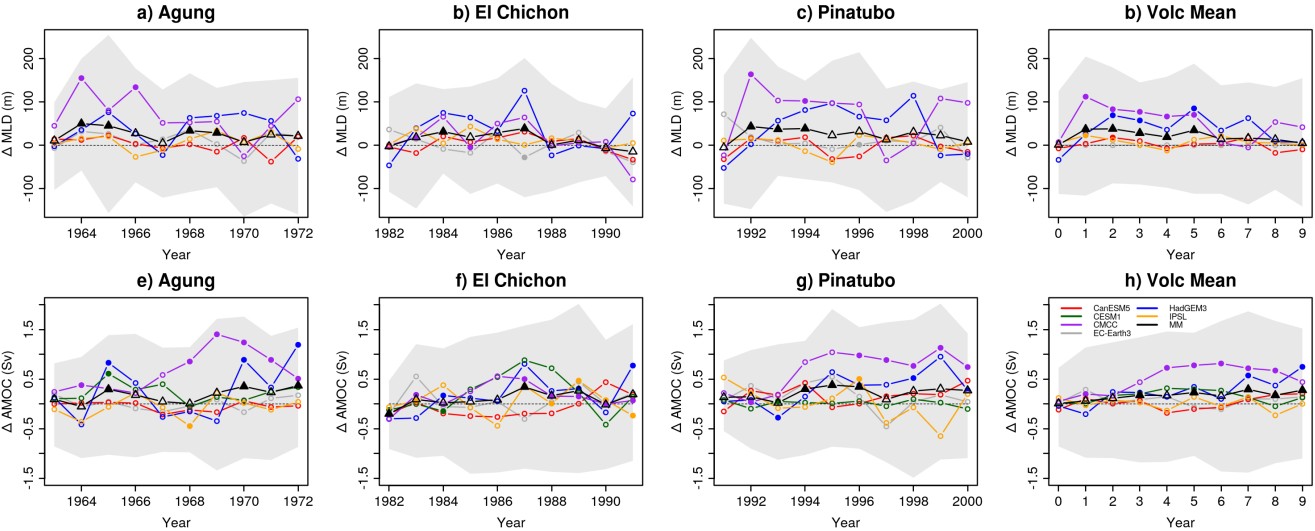

**Figure 16.** Volcanic response (DCPP-A minus DCPP-C) of the mixed layer depth in the Subpolar North Atlantic (50°N-65°N, 60°W-35°W) in February-March-April (top row) and the annual mean overturning stream-function averaged over 30°N-50°N and 1000m-2000m (bottom row). Filled circles/triangles indicate statistically significant differences (see methods). The ensemble mean for each model and the multi-model mean are shown. The shading is the multi-model member spread calculated as the 10th and 90th percentiles of the entire ensemble. The mixed layer depth for CESM1-1-CAM5-CMIP5 was not available and therefore could not be included.

densification of the upper ocean levels. The simulated increase in upper density erodes the mean stratification and is thus consistent with a subsequent enhancement in convection. As previously shown, this is indeed the case for the CMCC-CM2-SR5 and HadGEM3-GC31-MM models (Figure 16), for which the specific processes at play might differ as salinity changes
seem to dominate the density changes in CMCC-CM2-SR5, while for HadGEM3-GC31-MM temperature changes seem to be more important. The reason for the lack of responsiveness of the mixed layer depth in IPSL-CM6A-LR, despite the significant surface densification exerted by the volcanic forcing, is the very strongly stratified mean state density conditions that it presents in the region, which are linked to very fresh upper ocean conditions (Figure 17d-f). In the rest of the models no significant changes in temperature or salinity are observed in response to the eruptions, which explains why neither the mixed layer depth
nor the AMOC show a significant response.

## 4   Summary and Conclusions

Large volcanic eruptions can have significant climate impacts on seasonal-to-decadal timescales, some of which occur consistently across eruptions while others depend on aspects such as the magnitude, space–time structure of the volcanic aerosol concentrations, timing during the year and climate background conditions at the time of the eruption. Understanding these
commonalities and particularities in the responses, and to what extent they are model-dependent, is essential to make better





predictions should a new major volcanic eruption occur. The DCPP developed a specific protocol to improve our understanding of the effects of volcanic aerosols upon decadal prediction, which consists in repeating three sets of retrospective predictions initialised just before the eruptions of Agung (1963), El Chichón (1982) and Pinatubo (1991), but without the associated volcanic forcing (DCPP-C Boer et al., 2016). In this study we have analysed and compared these prediction sets with the baseline
predictions including all forcings in six CMIP6 decadal prediction systems.

All decadal prediction systems simulate a reduction in the global net TOA radiation fluxes, surface temperature and ocean heat content in response to the volcanic eruptions, with rather small inter-model differences in terms of the ensemble mean response. The magnitude of the eruption does influence the magnitude and persistence of the signals. The geographical pattern of the surface temperature response is also generally consistent across the models. For example, the first year following the
eruptions is characterised by a cooling of the Tropics and subtropics and a warming over the Eurasian Arctic sector, although the warming is not statistically significant for all eruptions. In later years, cooling spreads worldwide, with the strongest anomalies being found over the Arctic, with local cooling anomalies persisting for 5 to 9 years, depending on the eruption magnitude.

We have shown that there are some differences in the predicted radiative response among the three eruptions analysed. The eruption of Pinatubo was the largest and therefore it is reflected by simulating the strongest and most persistent anomalies in
TOA radiation fluxes, surface temperature and ocean heat content. The eruptions of Agung and El Chichón are weaker and of comparable intensity, but exhibit evident differences in the geographical distribution and temporal evolution of their forcings. While the eruption of Agung mainly affected the Southern Hemisphere, the eruption of El Chichón affected the Northern Hemisphere, something that is reflected in the TOA radiation and surface temperature anomaly patterns of the response. In contrast, the eruption of Pinatubo had a more meridionally symmetric response.

Besides the direct radiative cooling, the volcanic eruptions also excited dynamical responses. Since these responses are more sensitive to climatic noise they require larger ensembles to be detected, so we first analysed the multi-model and multi-eruption composite response, formed by 180 members. We note that this approach is useful to increase the ensemble size but can also mask some responses by including weaker eruptions (c.f. Bittner et al., 2016). The resulting composite response is characterised by a strong tropical warming in the lower stratosphere accompanied with a strengthening of the Northern Hemisphere polar
vortex in the first winter, which resembles a positive NAO-like pattern which is, however, not statistically significant. The ENSO response is characterised by the development of weak El Niño-like conditions in the first year after the eruption which then transitions to weak La Niña-like conditions in the second and third years. In the North Atlantic Ocean we have shown that there is a significant enhancement of the mixed layer depth in the Labrador Sea during the three boreal winters following the eruptions, and a weak but significant strengthening of the AMOC during years 2-9 after the eruptions. We have related these
responses to a reduction in density stratification in the Labrador Sea.

We have shown, however, that there are important differences in these dynamical responses, both across models and across eruptions. Multi-model composites for individual eruptions show that the acceleration of the Northern Hemisphere polar vortex only occurs in the eruptions of Agung and Pinatubo, while not for El Chichón. The lack of a response for El Chichón is probably related to a combination of factors, from its weak intensity, the geographical pattern of the forcing and the background climate
conditions. In the case of the ENSO response, we have shown that the for the eruptions of Agung and Pinatubo, the El Niño-like





state develops and peaks in the first year following the eruptions, while for the eruption of El Chichón the El Niño-like state occurs in the same year of the eruption. We have discussed that these differences are probably explained by the geographical pattern of the volcanic forcing (c.f. Pausata et al., 2020), the timing of the eruption and the ocean state (c.f. Predybaylo et al., 2020). We have also shown that there are important inter-model differences in these dynamical responses. For example, not all

models simulate an acceleration of the Northern Hemisphere polar vortex. The ENSO response is also model dependent since some models show a strong response and others remain unresponsive. Similarly, for the North Atlantic Ocean we have shown that the multi-model response comes exclusively from two of the models (CMCC-CM2-SR5 and HadGEM3-GC31-MM), which show coherent changes in Labrador Sea stratification, the mixed layer depths, and the AMOCs.

        To fully exploit these decadal hindcasts and determine whether including the volcanic forcing results in improved predictabil-
ity in these events, we have compared the predicted surface temperature in the three DCPP-A (with volcanic forcing) and the DCPP-C (without volcanic forcing) hindcasts with observations. This protocol also allows us identify and attribute observed variations to the volcanic forcing. We have shown that for the global mean surface air temperature, the DCPP-A hindcasts predict the observed anomalies significantly better by reproducing the post volcanic cooling. At the local scale, even though the volcanic forcing has a characteristic regional surface air temperature response pattern which evolves with forecast time, an
improvement in the DCPP-A hindcasts is only detectable for forecast years 2-5, when the volcanic signal is strongest. For other forecast times considered (year 1 and years 6-9), either the forecast error is greater than the volcanic impacts, the local volcanic signals are overwhelmed by internal variability and/or the regional response to the volcanic forcing is not correctly simulated by the models. In particular we have shown that the volcanic forcing seems to have a weak impact on ENSO, and in the case of Pinatubo degrades the predicted SST anomalies in the tropical Pacific Ocean, as shown in Wu et al. (2023). This is not the case
for the other two eruptions, which are no worse in the tropical Pacific with volcanic aerosols included. In contrast, in the North Atlantic Ocean, the volcanic forcing seems to be particularly important for reproducing the observed SST variability in the first few years following the eruptions. We also note that the hindcast corresponding with the eruption of Pinatubo is overall better at predicting the observed anomalies than for the eruptions of Agung and El Chichón. This could be because the eruption of Pinatubo had a stronger climatic impact and/or because the volcanic forcing is better constraint by the satellite observations
available.

        The results of this multi-model study provide further insight to the effect of volcanic eruptions on climate and predictability. We note however that further work is required to further understand several aspects. Since hindcasts are initialised from the observed state, disentangling the impacts from the volcanic forcing peculiarities and the background climate state is not always simple. The three volcanic eruptions in our hindcasts are insufficient to account for different background climate states, there-
fore idealised simulations as those proposed by VolMIP (Zanchettin et al., 2016, 2022) considering initial states based on the phase of different modes of climate variability should provide insightful results. Another source of uncertainty comes from the volcanic forcing itself (Toohey et al., 2014). For example, we have shown that CESM1-1-CAM5-CMIP5 has a considerably stronger tropical lower stratospheric temperature response than other models, which could be explained by the fact that it is the only model that did not use the CMIP6 forcing. This could indeed explain the differences in response in other variables
too. Understanding the sensitivity to the volcanic forcing is particularly relevant in a real-time climate prediction context, since

the Volcanic Response Plan (VolRES) following the next major eruption (a Stratosphere-Troposphere Processes and their role in Climate (SPARC) initiative, a core project within the World Climate Research Program (WCRP)) protocol consists in estimating the volcanic forcing of the future eruption using tools such as the Easy Volcanic Aerosol model (e.g. Toohey et al., 2016; Aubry et al., 2020). Finally, a limitation of the current DCPP-C protocol is that it is does not allow an assessment of the

impact of these volcanic eruptions on the forecast skill (as in Timmreck et al. (2016); Ménégoz et al. (2018); Wu et al. (2023)). Therefore, ideally future DCPP-C exercises could consider re-running all the hindcasts that overlap with these three volcanic eruptions in a way that would allow such analysis.

*Data availability.*    The data that support the findings of this study are openly available at the following URL/DOI https://esgf-node.llnl.gov/search/cmip6/.

*Code and data availability.*    For data retrieval, loading, processing and calculating metrics and visualisation, the startR (Automatically
Retrieve Multidimensional Distributed Data Sets. R package version 2.3.0. https://CRAN.R-project.org/package=startR, Manubens, Ho, Perez-Zanon and BSC-CNS, 2023) and s2dv (A Set of Common Tools for Seasonal to Decadal Verification. R package version 2.0.0. https://CRAN.R-project.org/package=s2dv, Ho, Perez-Zanon and BSC-CNS, 2023) R libraries have been used. The code to reproduce the results and figures presented in this study can be made available upon request.

*Author contributions.*    Data curation: ACH and MS. Formal analysis: RB. Investigation: all authors. Methodology: all authors. Visualization:
RB. Writing – original draft preparation: RB. Writing – review and editing: all authors.

*Competing interests.*    The authors declare that they have no conflict of interest.

*Acknowledgements.*    RB acknowledges support from the European Union's Horizon 2020 research project CONFESS No 101004156.



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





**Figure 17.** Multi-eruption mean response of the DJF Labrador Sea (50°N-65°N, 60°W-35°W) temperature, salinity and density profiles in years 1-3 after the eruptions (a-c) and absolute DJF Labrador Sea temperature, salinity and density profiles in years 1-3 after the eruptions in the predictions without volcanic forcing (d-f). Crosses indicate statistically significant differences (see methods). The ensemble mean for each model and the multi-model mean are shown. The shading is the multi-model member spread calculated as the 10th and 90th percentiles.