# Peer review of "Impact of volcanic eruptions on CMIP6 decadal predictions: A multi-model analysis"

_Earth System Dynamics, 2023_

## Author Comment (AC1)

**Reviewer 1:**

I read with interest the discussion paper by Roberto Bilbao and Colleagues on the impact of volcanic eruptions on CMIP6 decadal predictions. The study is interesting, the analysis overall sound and the manuscript is well written and organized. The manuscript is worthy of publication in ESD, also as a reference paper for the DCPP-A and -C experiments. I have just a main comment regarding abstract/conclusions and a few minor specific suggestions for the authors about some possible improvements to their manuscript.

Regarding my main comment on abstract and conclusions, it seems to me that, beyond the focus on prediction, the study revisits general aspects of volcanically forced climate variability. The latter analyses confirm previous results about, for instance, dependency of the response on magnitude and spatial structure of the forcing, the signal-to-noise ratio issue in detecting the response to moderate size eruptions, the lack of robustness in NAO response, and the tendency to produce a post-eruption El Nino response. My point is, several of the main messages in abstract and conclusions concern these aspects of volcanically forced variability, rather than volcanoes and predictions. This could still be viable, of course, but statements like "some differences across models and eruptions arise due to the varying magnitude and spatiotemporal structure of the volcanic forcing", or [decadal prediction systems show] "a strong agreement in predicting the radiative response to the volcanic eruptions, simulating a reduction in global mean top-of-atmosphere radiation fluxes, surface temperature" appear too general and vague, and, in my opinion, do not describe new advances in our understanding of the volcano-climate relationship. I feel that the paper would benefit from a narrower focus, in abstract and conclusions, on novel information about volcanic impacts on decadal predictions that emerges from this approach/analysis: In the abstract, this key part appears mainly in the statement that "including the volcanic forcing results in overall better predictions" and in the last sentence. I recommend the authors to consider revisiting these important parts of their paper.

Reply: Thanks for the very relevant suggestion. We agree that neither the abstract nor the conclusions were clearly separating which results confirm previous findings on the climatic impacts of volcanic eruptions and which ones provide new insights. We have re-written both sections to make this more explicit, discussing the results in the context of previous studies, and highlighting the particular interest in using decadal predictions to explore the sensitivity to the volcanic forcing.

Some specific comments:

Line 182-185: It is worth comparing these results with Zanchettin et al. (2022): results from volc-pinatubo-full indicate "maximum expected cooling ranging across models between about -0.27 and -0.38 and a multi-model mean of about -0.33". The good overlap suggests minor influence from initial states and background conditions?

Reply: Interesting point, we have added a sentence about this: 'The good overlap between the temperature anomalies reported here and found in Zanchettin et al. (2022) (multi-model mean of about -0.33°C) for the eruption of Pinatubo, suggests that the influence from the initial conditions and the background state is small, at least for the global mean.'

Line 240-241: aren't points (1) and (2) different flavors of the same explanation?

Reply: We agree that these two points create confusion so we have simply removed the first one. This has been corrected in the manuscript.

Section 3.5: a deeper discussion about the implications of the constructive/destructive superposition mechanism by Swingedouw et al. (2015) might be worthy here.

The interference mechanism proposed in Swingedouw et al. (2015) has different ingredients, and notably the need for long simulations covering the last three volcanic eruptions, which is not the case here with decadal predictions of 10 years. Also, the analysis is focusing on the **differences** between simulations with or without a volcanic eruption, which also differ from Swingedouw et al. (2015) that did not look at differences but "absolute" values of the AMOC evolution. Still, what the present study can provide is an estimation of the magnitude of AMOC response to the different eruptions, but 10 years remains a short time frame since the maximum of the response was found more than 10 years after the eruption in Swingedouw et al. (2015). Thus, the experimental design is not allowing an in-depth discussion of this, mainly because the simulations analysed are too short. Therefore, we only slightly further discuss this mechanism in the manuscript, and notably the fact that the AMOC is actually increasing following an eruption, with larger magnitude for stronger eruptions.

Line 469: typo (is does)

Reply: Corrected.

Figure 5: could you extend the plots to encompass the whole hindcast period? At least for El Chichon observations appears to be in a colder anomalous state than hindcasts: does this discrepancy hold since the beginning? What if you use the pre-eruption period as baseline to calculate the anomalies (e.g., the five years before each eruption)? This comment stands also for the SSTs in Figure 14.

Reply: We have recomputed Figure 5 in the paper but including the year before each eruption (Figure 1). In the case of El Chichón, observations no longer appear to be in an anomalous cooler state. This figure shows that the hindcasts encompass the observations rather well.

Regarding the idea of using a different baseline period, since these are decadal predictions, the anomalies are computed using the standard approach of subtracting a forecast time dependent climatology in order to correct for the forecast drift. If we consider computing the anomalies with respect to the five years before the eruptions, this means that only five hindcasts are used to estimate the forecast drift, which might not be efficient to characterise it. Also, note that this would not be possible for the eruption of Agung, since the first predictions in our hindcast start in 1960. The recent study of Meehl et al. (2022) has explored other methodologies such as computing the anomalies of a forecast with respect to the previous 15 years, that is, using the previous 15 start dates, but it has not shown major benefits in terms of detecting improvements in predictive skill.

Meehl, G.A., Teng, H., Smith, D. *et al.* The effects of bias, drift, and trends in calculating anomalies for evaluating skill of seasonal-to-decadal initialized climate predictions. *Clim Dyn* 59, 3373–3389 (2022). https://doi.org/10.1007/s00382-022-06272-7

[Figure]

Figure 1. Monthly mean global near-surface temperature anomalies (◦C) of the predictions initialised in 1962, 1981 and 1990 for the DCPP-A (with volcanic forcing) and DCPP-C (without volcanic forcing) experiments. HadCRUT5 is used as the observational reference (dashed line). The anomalies have been computed with respect to the period 1970-2005 (see methods for further information). The shading is the multi-model member spread calculated as the 10th and 90th percentiles of the entire ensemble.

Regarding the NAO and ENSO responses: analysis of volc-pinatubo-full experiment yields overall similar conclusions to this study regarding the lack of NAO and, to a lesser extent, ENSO response to the Pinatubo eruption, using the analog approach of paired anomalies, but with some noticeable differences (for instance regarding ENSO response in IPSL-CM6A-LR. Maybe it is worth commenting on this?

Reply: Interesting point. We have added a comment to highlight the difference we find in IPSL-CM6A-LR: 'Note that the results for the IPSL-CM6A-LR model contrast with those of Zanchettin et al. (2022), where it is shown that in the idealised Pinatubo experiments (volc-pinatubo-full experiment) this model does simulate the El Niño/La Niña responses.'

---

## Author Comment (AC2)

**Reviewer 2:**

This article presents results from a new set of decadal forecast models which allows the authors to isolate the effect of the three most recent large volcanic eruptions on climate. The analysis is interesting and very clearly presented, the paper is well written and methodology seems sounds – so I think that this article would be worthy of publication is in this journal.

I have two major concerns.

The first is due to the small sample size (only three eruptions) and the comparison to the observations. While the large ensemble sizes allow for proper verification of model results, it is not clear to me how best to interpret the comparison to observations. It the signal to noise ratio is low (as it presumably is for regional results). Including a volcano could improve skill in general while degrading the fit to certain aspects of climate in a specific case. Equally the models could have no skill while by chance a better fit to observations could occur. I think more discussion regarding this is needed in the context of predictability, so that the conclusions are clear and the results are not over interpreted.

Reply: We agree with the reviewer that while three volcanic eruptions is a small sample size, we are limited by the observational record. Despite the small sample size, previous studies have shown the improvement (and deterioration) of the predictive skill associated with the volcanic forcing based on retrospective forecasts for the last several decades (e.g. Timmreck et al. (2016), Ménégoz et al. (2018), Wu et al. (2023)). With our protocol for three selected initialization dates right before the eruptions, we cannot estimate the impact of the volcanic forcing on the predictive skill, so our approach has been therefore to perform a qualitative comparison of the two alternative forecasts (with and without volcanic forcings) with the observations, to see in which cases adding the volcanic forcing improves the agreement with observations. The analysis of the DCPP-A and DCPP-C hindcast differences motivates such comparison since the post-volcanic cooling is significant and therefore should be detectable in the observations. We agree with the reviewer however that some of these apparent improvements, in particular those related to dynamical variables, might occur by chance, a caveat that we now explicitly mention in the conclusions sections.

The second is regarding novelty. Certainly, all findings do have novelty as they represent results from a new set of model simulations. However, much of this article details changes in aspects of the climate by volcanic eruptions which are already well documented in many other studies - ENSO, NAO, winter warming, AMOC etc as outlined in the introduction. I therefore think that more should be added to put them into the context of previous studies. What is new and what confirms or contradicts existing work?

Reply: We think that in the results section we have included many references to previous studies contextualising the results, but we agree that in the abstract and conclusions section we should have been more precise and state which results support previous studies and which results are new. We have re-written both to improve these aspects and insist on the fact that we are in a predictive mode, which differ from a number of former studies about volcanic eruption impact

Minor points:

Line 127-128. "on" the 30$^{th}$ October in both cases

Reply: Corrected.

Paragraph L130. Would it be possible to include some discussion about the uncertainty in the volcanic forcing. Agung in particular is pre-satellite, so are the uncertainties larger? I note that the reference given only seems to covers the satellite era.

Reply: There is no peer-reviewed publication documenting the CMIP6 volcanic dataset, and therefore no detailed description of such uncertainties. We have nonetheless cited the very short documentation provided by Luo (2018), who prepared this volcanic forcing dataset. Within this documentation, Thomason et al. (2016) is referenced for further details. We do indeed expect the uncertainties in the volcanic forcing of Agung to be larger since before 1979 the forcing dataset was produced by the AER-2D model (Arfeuille et al., 2014). While it is important to consider this uncertainty when making conclusions, it is beyond the scope of this study to quantify it. We have included a comment in the paragraph.

Line 143 – could you give more information regarding the ozone depletion as this isn't that clear to me.

Reply: Previous studies have shown that changes in stratospheric circulation and temperature in response to volcanic aerosols can affect stratospheric ozone photochemistry, and therefore result in ozone changes (e.g. Stenchicov et al., 2002; and references within). We simply note that these processes which lead to ozone depletion are not simulated in these climate models. Also, note that climate-chemistry models have not yet been used in climate prediction.

L155-156 – were these the infilled version of HadCRUT5?

Reply: Yes, it is the HadCRUT5 analysis gridded data ensemble mean.

Figure 8 – could you specify what period the anomalies are respect to?

Reply: It is stated in the figure caption: 'The anomalies have been computed with respect to the period 1970-2005.'